# The Flow of Green Exercise, Its Characteristics, Mechanism, and Pattern in Urban Green Space Networks: A Case Study of Nangchang, China

Zhenrao Cai [1], Dan Gao [1,2], Xin Xiao [1], Linguo Zhou [1] and Chaoyang Fang [1,2,3,*]

[1] School of Geography and Environment, Jiangxi Normal University, Nanchang 330022, China; 202050000004@jxnu.edu.cn (Z.C.); gisxx@jxnu.edu.cn (X.X.)
[2] Key Laboratory of Poyang Lake Wetland and Watershed Research, Ministry of Education, Jiangxi Normal University, Nanchang 330022, China
[3] Nanchang Base, International Centre on Space Technologies for Natural and Cultural Heritage (HIST) under the Auspices of UNESCO, Nanchang 330022, China
* Correspondence: fcy@jxnu.edu.cn

**Abstract:** An urban green space network provides safe and green exercise routes for residents. This study selected Nanchang as the study area. Using fitness application data, we explored the mobility of people exercising in the network, i.e., the flow of green exercise. Spatial analysis based on social networks, GIS, and the gravity model was used to analyze the nodes, network characteristics, and mechanism of the flow of green exercise. The results show that there were differences in the hierarchy and attractiveness of nodes. Distance had an important influence on green exercise. It was found that walkers moved between adjacent parks, runners visited more parks and corridors, while cyclists covered longer distances and preferred to explore suburban green spaces. The length of the exercise route in green spaces had a positive effect on mobility. Because of the many combinations of patches and corridors, three flow subnetworks were formed. In addition, the green space network expanded the scope of exercise services in the central node. The management of green spaces should pay attention to the social value of urban green space networks and create a hierarchical and interconnected green space for exercisers.

**Keywords:** urban green space network; green exercise; social network; ecosystem services; patch–corridor

## 1. Introduction

Urbanization negatively affects people's livelihoods because it spreads sedentary behavior, such as spending a lot of time seated at work [1], watching TV [2], etc. Physical inactivity is the main factor that causes increased global morbidity and premature death. It accounts for 6% of deaths globally. In addition, 1.4 billion adults are at risk of disease due to physical inactivity [3]. The 2020 World Health Organization guidelines recommend limiting sedentary behavior and emphasize the importance of intense activities [4]. Green spaces provide a variety of ecosystem services, such as temperature regulation, rainwater management, waste disposal, air quality regulation, pollination, recreation, and aesthetic appreciation [5,6]. One of its most important aspects is the exercise service [7]. It provides spaces for residents to exercise and allows them to keep exercising [8]. Exercise in nature, specifically urban green spaces, is called green exercise [9]. This form of exercise can enhance the benefits of physical activity and have a positive impact on human health [10,11]. Therefore, urban planners and policymakers should provide an attractive urban green space for urban residents to exercise in. Recent studies have found that the differences in the characteristics of urban green spaces directly affect green exercise [12]. It is necessary to explore more characteristics of urban green spaces and the factors that affect green exercise.

Previous studies have focused on the impact of internal environmental characteristics of a single park and its use [13–16], including park facilities, convenience, security, size, and

proximity. These studies mainly conducted questionnaire surveys [14] and GPS equipment tracking [16] to explore the impact of urban green space characteristics on the amount and intensity of exercise. With the development of big data, researchers began using social network check-in data, such as from Weibo [17] and Twitter [18], population density map data [19], and mobile phone signaling data [20], to see how urban residents access and use parks. They also used trajectory data from fitness applications to analyze the space for exercising in macro-scale urban park systems, such as the trajectory data of jogging [21] and walking [22]. Although there is plenty of research on this subject, there are still some gaps that need to be filled. At present, most studies focus on the visits of urban residents to parks and few of them are limited to visualizing exercise activities and popular parks. Most of these studies tend to describe static phenomena rather than explore processes and patterns.

Due to the construction of green networks [23] and eco-garden cities [24] in major cities in China, the number and density of urban green spaces has improved. The first law of geography holds that objects near one another are more related than distant objects [25]. When the spatial scale and attractiveness of parks are high, a flow between parks occurs. Urban green spaces do not only provide animals opportunities to migrate, reduce pollution, cool the air, reduce noise, and foster social interaction, but also enable urban residents to exercise and enjoy leisure activities. In recent years, some people have been using the green space network as an exercise space in which they perform popular short- to long-distance green activities such as walking, jogging, running, and cycling. The process of exercising in one green space and then another is essentially a process of visiting ecosystem services. It represents a type of ecosystem service flow named "people to nature" [26]. The flow contains important spatial information, such as where the exercise takes place, the order of exercise, and the relationship between exercises and green spaces. However, research on the spatial relationship between green spaces is still insufficient. There is a need to understand the process of using exercise services in the urban green space network, which this study called the flow of green exercise. The flow is a spatiotemporal process in which exercisers enjoy exercising in multiple green spaces; that is, exercisers flow from one green space to another while exercising [27]. The flow is necessary to ensure fair and sustainable use of the ecosystem. This study considered walking, running, and cycling as dominant activities performed in green spaces. Because of their large capacity, low cost, high precision, and open browsability, China's fitness applications such as Keep, Codoon, Xingzhe, 2bulu, and Joyrun (similar to Strava) record and share a large amount of track data. A large number of movement trajectory data can provide reliable sources for studying the flow of green exercise in urban green space networks.

This paper selected Nanchang as the study area to solve the following problems: (1) What are the characteristics of green space nodes and networks under the flow of green exercise? (2) What is the mechanism of this flow? Does the flow network produce different spatial patterns due to the location of nodes and the functional relationship between them? How does the flow relate to the nodes? This paper aims to understand the flow of green exercise by answering these questions, effectively support the design and management of green infrastructure and vacant parks, and provide a basis for constructing routes for outdoor activities in urban areas on a large scale.

## 2. Study Area, Materials and Methods

### 2.1. Study Area

This study chose the eastern part of the main urban area of Nanchang (28°34′ N–28°46′ N, 115°49′ E–116°7′ E) as the study area. Nanchang is located near Poyang Lake in the middle and lower reaches of the Yangtze River. The city has built a series of urban green spaces around the river and lakes. This study excluded fee-paying scenic spots, zoos, and smaller parks (less than 0.03 km²). Thus, 31 green spaces in the main urban area were selected for analysis (Figure 1). Since 2020, several city-level landscape avenues with

walkways and cycling lanes have been built in Nanchang. These landscapes are close to one another due to interconnected green spaces.

**Figure 1.** Distribution of green spaces in the study area.

*2.2. Data*

The boundary data of the green space was obtained from the planning data issued by the Nanchang Municipal People's Government [28]. The 31 green spaces include park green spaces, scenic green spaces, and waterfront green spaces under ecological protection. They are free and open spaces managed by the government. These 31 workout destinations can be found in exercise recording software because many people exercise in them. Based on the landscape shape index [21] of the green spaces and the "patch–corridor–matrix" model, the green spaces were divided into two types: patches and corridors (Table 1). Corridors are narrow and long, mostly along river channels, while patches are wider and open spaces, such as fields.

**Table 1.** Classification of green spaces.

| Node Type | Specific Name |
| --- | --- |
| Patch | 1. Aisihu Park, 2. Xianghu Park, 3. Xianghunan Park, 4. Nantanghu Park, 5. Nanaixihu Park, 6. Yaohu Park, 7. Yuweizhou Park, 8. Jinsha Park, 9. People's Park, 10. Bayi Park, 11. Xianshihu Park, 12. Ruziting Park, 13. Chaoyangzhongyang Park, 14. Qingshanhu Sports Park, 15. Changnan Sports Park, 16. Baojia Sports Park, 17. Qingshanhu Scenic Spot, 18. Meihu Scenic Spot, 19. Qingyun Reservoir |
| Corridor | 20. Ganjiangdong Scenic Belt, 21. Yaohubei Scenic Belt, 22. Ganjiangbei Scenic Belt, 23. Yudaihezong Channel, 24. Yudaihexi Channel, 25. Yudaihedong Channel, 26. Chaoyang Greenway, 27. Xiongxihe Greenway, 28. Taohuahe Park, 29. Yudaihebeizhi Park, 30. Fuhe Park, 31. Yuhu Park |

The movement trajectory data were obtained from the five most popular fitness applications in China, of which Keep, Codoon, Xingzhe, and 2bulu record three types of exercise, walking, running, and cycling, while Joyrun only records running. The five applications provide users with display and share modules so that they can share and

upload their activities. The data in these modules are free to access. This study collected the exercise trajectory attribute data based on the chronological order of the uploads. The data are from January 2017 to November 2021. The collected exercise attribute data include exercise routes, time of day, coordinates of the starting points, and the names of the green spaces that people used for exercising. To guarantee data anonymity, information such as age and gender was not collected. The route data were collected in the following way. Firstly, a person must enter two or more green spaces in one day and exercise for a specific amount of time in the green spaces. A complete trajectory distance represents one route that can have multiple paths. As shown in Figure 2, the total distance from green space A to green space B, B to C, and C to D was counted as one route. Notations (1), (2), and (3) represent direct flow paths, which show a direct relationship between the nodes, while (4), (5), and (6) represent indirect flow paths, which show an indirect relationship between the nodes. Figure 2a represents a network diagram of green exercise routes in four green spaces. Based on this diagram, an asymmetric matrix was constructed (Figure 2b). Moreover, if an exercise route is beyond the scope of the study area, the route was not included in the study. Lastly, only the latest trajectory attribute data of an exerciser were collected if the exerciser frequently repeated exercise on a route. However, if the person used multiple exercise routes, the data from each route were collected once. This study used the matrix data to analyze the characteristics of the flow of green exercise.

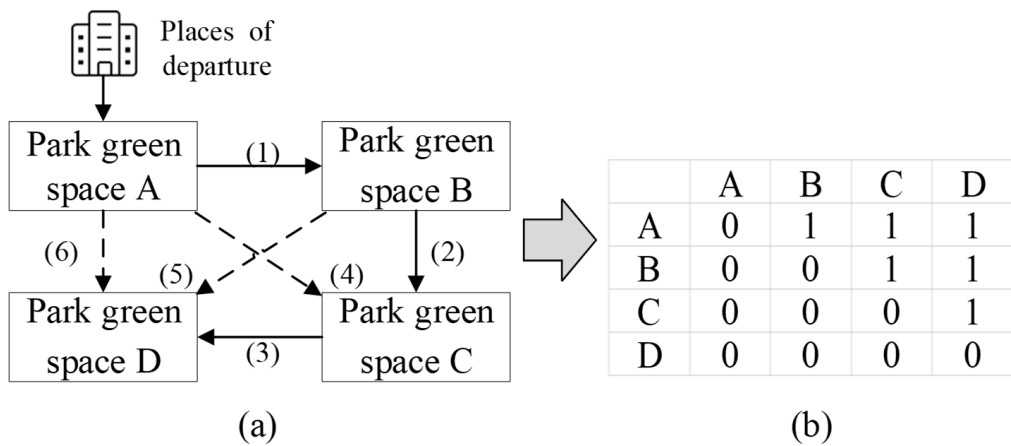

**Figure 2.** The relationship between the green spaces and their nodes: (**a**) the network diagram; and (**b**) the network matrix.

In this study, a total of 1031 users exercised on 1649 green exercise routes. These routes were transformed into an original matrix table of 31 rows × 31 columns. Furthermore, this study recognized the green spaces as nodes, while the flow network consisted of multiple paths. Thus, the matrix had 31 nodes and 267 paths. The total flow (total number of paths) of exercise was 2613, of which 217 was walking flow, 1383 was running flow, and 1013 was cycling flow, accounting for 8%, 53%, and 39% of the total flow, respectively.

This study identified two types of exercisers. Type A are exercisers who move through the urban green space network to reach their target node, while type B are the exercisers who directly reach their target node without moving through other green spaces. In addition, the starting point data of the two types of exercisers were obtained from the applications. The standard for selecting starting point sample data was that one end of the trajectory data must be located in a residential area or on industrial or commercial land. If an exerciser performs two types of exercise, data for each type were collected once. Based on the display module, this study collected the starting point data of 2571 users at three first-level patches in order of the time of upload. The starting point sample data of type A was 670.

*2.3. Methods*

2.3.1. Method for Evaluating Nodes and Network Characteristics

Previous studies understood the spatial network in terms of spatial entities and used space syntax and graph theory to explore the ecological network structure of urban green spaces [29,30]. Social network analysis is a quantitative analysis developed by sociologists and is based on mathematical methods and graph theory. In recent years, different types of research have been conducted in sociology, economics, management, and other fields, helping integrate the relationship between individuals in the network, micro-network, and network structure [31]. Due to the availability of tourism flow data, social network analysis is mostly used to analyze the flow of tourists in scenic spot networks [32]. This study, however, used social network analysis to explore network structure based on the flow of exercise. Likewise, it used the social network analysis software UCINET to calculate the centrality of a node and network and the density of a network [33]. The following equations were used to calculate the weighted average distance of the outflow and inflow of exercisers at each node:

$$D_{outi} = \sum_{j=1}^{n} I_{ij}d_{ij} / \sum_{j=1}^{n} I_{ij} \tag{1}$$

$$D_{ini} = \sum_{j=1}^{n} I_{ji}d_{ij} / \sum_{j=1}^{n} I_{ji} \tag{2}$$

where $D_{outi}$ and $D_{ini}$ are the weighted average distances of the green exercise outflow and inflow in green space $i$. $I_{ji}$ is the flow from node $j$ to node $i$, $I_{ij}$ is the flow from node $i$ to node $j$, and $d_{ij}$ is the distance between the two nodes. Based on network analysis of the vector data of road networks, this study calculated the distance from one node to the other. Finally, $n$ is the number of nodes in the network.

2.3.2. The Mechanism behind the Flow Based on the Gravity Model

Spatial flow is an external characteristic of the spatial interaction between two places and an important concept in geography. There are many mature measurement methods for it. In interaction studies, the most important one is the gravity model [34]. The equation is the following:

$$G_{ij} = KP_i^a P_j^y d^{-\beta} \tag{3}$$

The natural logarithms on both sides of Equation (3) are expanded and added to the multiple linear regression equation:

$$lnG_{ij} = lnk + alnP_{in} + ylnP_{jn} - \beta lnd_{ij} \tag{4}$$

where $G_{ij}$ is the intensity of the interaction between $i$ and $j$, $P_i$ and $P_j$ are the attractiveness of $i$ and $j$, $d$ is the distance between $i$ and $j$, $k$ is a constant term, and $\alpha$, $\gamma$, and $\beta$ are regression coefficients. In this study, the flow between two nodes ($G_{ij}$, the sum of $l_{ji}$ and $l_{ij}$) was used to represent the intensity of the interaction between $i$ and $j$. In the actual calculation, $P_i$ and $P_j$ are often replaced by the area size of two places. Not only does the length of the exercise route inside the green space represent a basic requirement for activities to take place and enables the person to move from one space to another, but it also represents an important factor that attracts people. Therefore, the length of the main exercise route of the outflow place $i$ and inflow place $j$ represents $P_i$ and $P_j$ in this study, respectively. $d$ is the actual distance between the two nodes. When $G_{ij} \geq 3$, the authenticity of flows is maintained, while accidental flows are excluded. Therefore, the path with $G_{ij} \geq 3$ represents the research object in this study. The Shapiro–Wilk/Kolmogorov–Smirnov tests, histograms, and Q–Q graphs were conducted on $lnG_{ij}$ including three types of exercisers. The results showed that the logarithms of the cycling and running flow had normal distribution. Due to little sample data on the walking flow, this study only analyzed the mechanism behind the running and cycling flows. In this study, the variance inflation factor (VIF) of each variable

was calculated to test the multicollinearity between the variables. If the VIF value is greater than 10, appropriate processing is required. In the running and cycling flow, it is expected that the length of the exercise route significantly improves the size of the flow, while spatial distance restrains it.

2.3.3. Measuring the Spatial Spillover Effect of Green Space Networks

The standard deviational ellipse analysis tool in the ArcGIS software represents a geostatistical method that can accurately reveal the spatial distribution characteristics of various geographical elements [35]. By analyzing the elliptical distribution of standard deviations (changes in direction, area, and longitudinal axis) of the starting point data of type B exercisers and that of both types of exercisers (the sum of A and B exercisers), this study explored the influence of green space networks (by adding type A exercisers) on the service scope of some nodes.

**3. Results**

*3.1. Analysis of Node Characteristics*

3.1.1. Analysis of Node Centrality in the Total Flow

This study divided the nodes into three levels. Generally speaking, the higher the level, the smaller the total number of nodes. In this study, the distribution ratio of the number of nodes in the three levels is 3:5:8 (Table 2).

**Table 2.** Network characteristics under different exercise scenarios.

| Flow Type | Number of Nodes | Number of Paths | Network Density | Network Centrality | Average Flow of a Path | Proportion of the Flow |
|---|---|---|---|---|---|---|
| Walking flow | 29 | 82 | 0.101 | 0.117 | 3.56 | 8.30% |
| Running flow | 31 | 156 | 0.168 | 0.335 | 12.81 | 52.93% |
| Cycling flow | 28 | 193 | 0.255 | 0.227 | 8.37 | 38.77% |
| Total flow | 31 | 267 | 0.287 | 0.214 | 16.13 | 100.00% |

This study calculated the centrality of each node in the total flow (the sum of the direct and indirect flows) (Figure 3). According to the results, the centrality of nodes was related to the size of patches and the length of corridors. The centrality of Qingshanhu Scenic Spot, Xianghu Park, and Aisihu Park ranked in the top three, reaching 457, 425, and 419, respectively, due to their large size and high popularity. These nodes were followed by the Yudaihezong Channel, Fuhe Park, and Ganjiangdong Scenic Belt, which are long riverside recreation belts with a centrality of 405, 275, and 164, respectively. The smaller patches, such as Ruziting Park, and the shorter corridors, such as Taohuahe Park, showed low centrality. The spatial position of a node also affected its centrality. For example, the size of Bayi Park is smaller than that of Nantanghu Park. However, the former had a higher centrality because it is located in the city center and is more populated. The length of Yudaihebeizhi Park was shorter, but its centrality was higher (98). This corridor is located in the city center, such as Bayi Park, and is also close to the large Qingshanhu Scenic Spot.

According to the centrality of the total flow, the nodes were divided into three levels. Nodes with a centrality greater than 160 represent the first level. This level includes three patches and three corridors, which are the most important areas for green exercise. The second level constitutes five patches and five corridors based on a centrality between 60 and 160. They are distributed on the periphery of the first-level nodes. In this study, first-level and second-level nodes were collectively referred to as primary nodes. The third level consists of eleven patches and four corridors based on a centrality between 5 and 60. These nodes are marginal nodes.

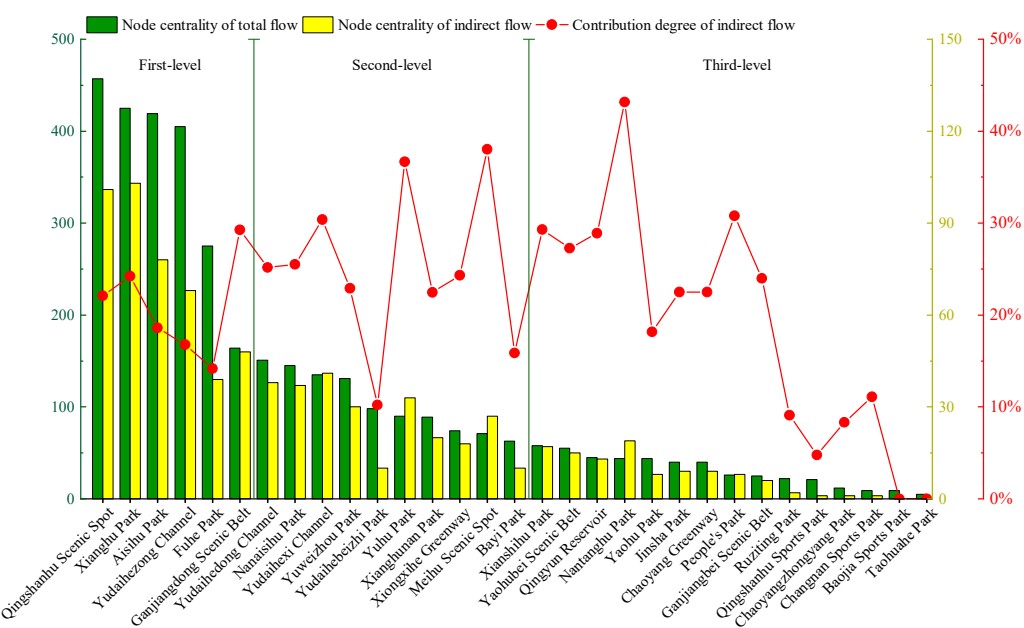

**Figure 3.** Centrality of each node in the total and indirect flows.

### 3.1.2. The Effect of the Indirect Flow on the Centrality of the Nodes

The total centrality of nodes in the indirect flow was 810, accounting for 22.2% of the total centrality of nodes in the total flow. The indirect flow had a certain complementary effect on the total flow. In terms of growth, there was a positive correlation between the centrality of nodes in the indirect and total flow. In general, the higher the centrality of a node in the indirect flow, the higher the centrality of the node in the total flow (Figure 3). The indirect flow showed whether long-distance nodes were attractive to exercisers. The centrality of first-level nodes in the indirect flow was higher, which means that they were more attractive than long-distance ones, so more people entered them through other nodes. In contrast, the centrality of third-level nodes in the indirect flow was generally low as they were the least attractive. The contribution of the indirect flow represents the ratio of centrality between each node in the indirect to that in the total flow. Based on the contribution, the indirect flow had a greater effect on some second-level nodes. The contribution of the indirect flow of Nantanghu Park, Meihu Scenic Spot, and Yuhu Park was more than 35%, indicating that these spaces were very attractive and play an important role in the flow network.

### 3.2. Flow Network Analysis

### 3.2.1. The Flow Network under Different Types of Exercise

The results of flow networks analysis under different types of exercise are shown in Table 3 and Figure 4. It was found that the exercises affected network density and the shape of the flow. The number of nodes and paths in the walking flow was 29 and 82, respectively. In the walking flow, network density and centrality were the lowest, 0.101 and 0.117, respectively. The low density and centrality were related to the characteristics of walking. This indicates that walkers tend to flow between adjacent nodes, making the network flat. The number of nodes was the highest (31) in the running flow. Likewise, network centrality was the highest (0.323) in the running flow, indicating that runners tend to move towards first-level nodes. Because of this, agglomeration of the running flow was the most significant compared to other flows. The number of paths and network density was the highest in the cycling flow, which is related to the cyclists' ability to easily travel long distances from one node to another. Moreover, network centrality of the cycling flow was lower (0.227) than that of the running flow (0.335). In the suburban parts of Yaohubei Scenic Belt, Yuweizhou Park, and Meihu Scenic Spot, the number of paths and nodes in the

cycling flow was greater than that of the number of paths in the running flow (Figure 4), indicating that cyclists are better at exploring green spaces. Therefore, walkers tend to move between nodes that are close to each other, runners tend to move towards first-level nodes, while cyclists tend to move towards nodes with different degrees of distance between them.

**Table 3.** Network characteristics under different flow volumes.

| Flow Volume Threshold | Number of Nodes | Number of Paths | Network Density | Network Centrality | Average Flow of a Path | Proportion of the Flow |
|---|---|---|---|---|---|---|
| ≥3 | 30 | 142 | 0.163 | 0.273 | 24.39 | 97.09% |
| ≥12 | 24 | 53 | 0.092 | 0.355 | 44.06 | 85.99% |
| ≥24 | 16 | 25 | 0.104 | 0.093 | 71.65 | 71.30% |
| ≥48 | 10 | 12 | 0.133 | 0.145 | 114.92 | 52.77% |

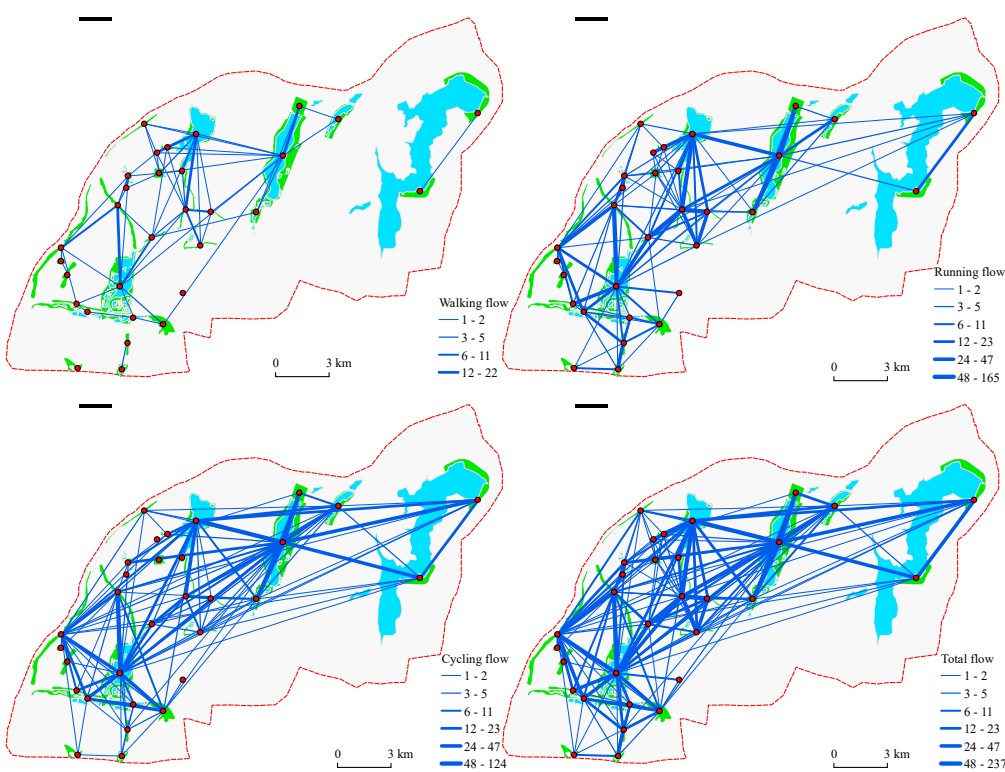

**Figure 4.** Flow networks under different exercise types.

### 3.2.2. The Flow Network under Different Flow Volumes

The flow volume between nodes was different, so the characteristics of the network under different flow volumes were different, too. According to the difference in flow volume between nodes, the total flow was divided into four levels: $G_{ij} \geq 3$, $G_{ij} \geq 12$, $G_{ij} \geq 24$, and $G_{ij} \geq 48$ (Table 3 and Figure 5). With the increase in the flow volume, the number of nodes, the number of paths, and the proportion of the flow decreased. The flow between two nodes increased with the increase in the flow volume. With the increase in $G_{ij}$, network density first decreased and then increased. However, there was no obvious relationship between network centrality and the change in the flow volume. When $G_{ij} \geq 3$, the proportion of the total flow reached 97.09%, dominating the flow network and constituting the basic structure of the flow network in the green space. As can be seen from Figure 5, with the increase in flow volume, primary nodes and main flow paths appeared. When $G_{ij} \geq 48$, there was still a large flow between adjacent first-level nodes, such as Qingshanhu Sports Park and Aisihu Park, while the flow between adjacent third-level nodes, such as Bayi Park and Ruzi Pavilion, stopped. This shows that large green spaces

with larger distances between them have higher interconnection than small green spaces close to each other.

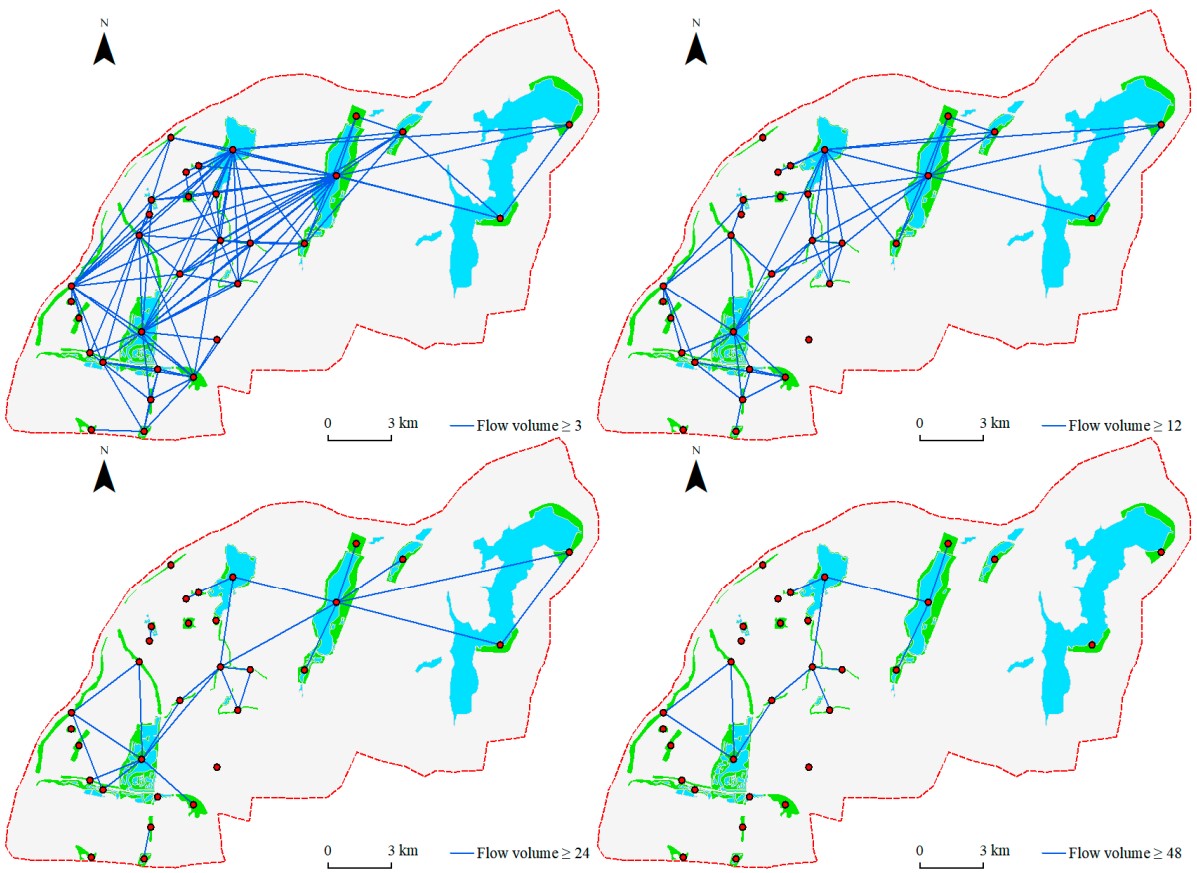

**Figure 5.** The flow network under different flow volumes.

### 3.2.3. The Flow Network under Different Distances

Distance is an important factor that affects flow. This study further explored the influence of distance on network characteristics. The smaller the flow between nodes, the greater the contingency of the relationship between the nodes. To better analyze the influence of the distance, it is necessary to define a control distance. In this study, distance is divided into four categories: 0–2 km, 2–4 km, 4–6 km, and ≥6 km. The influence of distance on flow network is analyzed using the numerical step change.

With the increase in distance, the number of nodes, the number of paths, and the flow between nodes gradually decreased. In this case, other indicators had no obvious relationship with the change in distance (Table 4). Generally speaking, distance had an important influence on the flow of green exercise and the law of distance attenuation was obvious. The larger the distance between nodes, the smaller the flow between them. When the distance was 0–2 km, the average flow of a path was 22.87, indicating that short-distance flow dominated the flow of green exercise. However, there was a great difference between nodes at the same distance in the flow. For example, the flow between Qingshanhu Scenic Spot and Yudaihezong Channel was much larger than that between Yudaihebeizhi Park and Xianshihu Park, indicating that the intensity of the spatial connection between larger green spaces was greater than that between small ones. As can be seen from Figure 6, when the distance was 0–4 km, the flow volume of some paths was very large (more than 48). When the distance was greater than 6 km, there were still some paths with a large flow volume (greater than 24), indicating that distance is not the only factor that influences the flow in the urban green space network.

**Table 4.** Network characteristics under different distances.

| Distance | Number of Nodes | Number of Paths | Network Density | Network Centrality | Average Flow of a Path | Proportion of the Running Flow | Proportion of the Cycling Flow |
|---|---|---|---|---|---|---|---|
| (0, 2] | 26 | 77 | 0.119 | 0.236 | 22.87 | 61.39% | 28.73% |
| (2, 4] | 19 | 27 | 0.079 | 0.098 | 14.11 | 41.21% | 54.86% |
| (4, 6] | 17 | 21 | 0.077 | 0.037 | 7.52 | 39.87% | 55.70% |
| ≥6 | 12 | 17 | 0.129 | 0.246 | 8.53 | 9.66% | 88.97% |

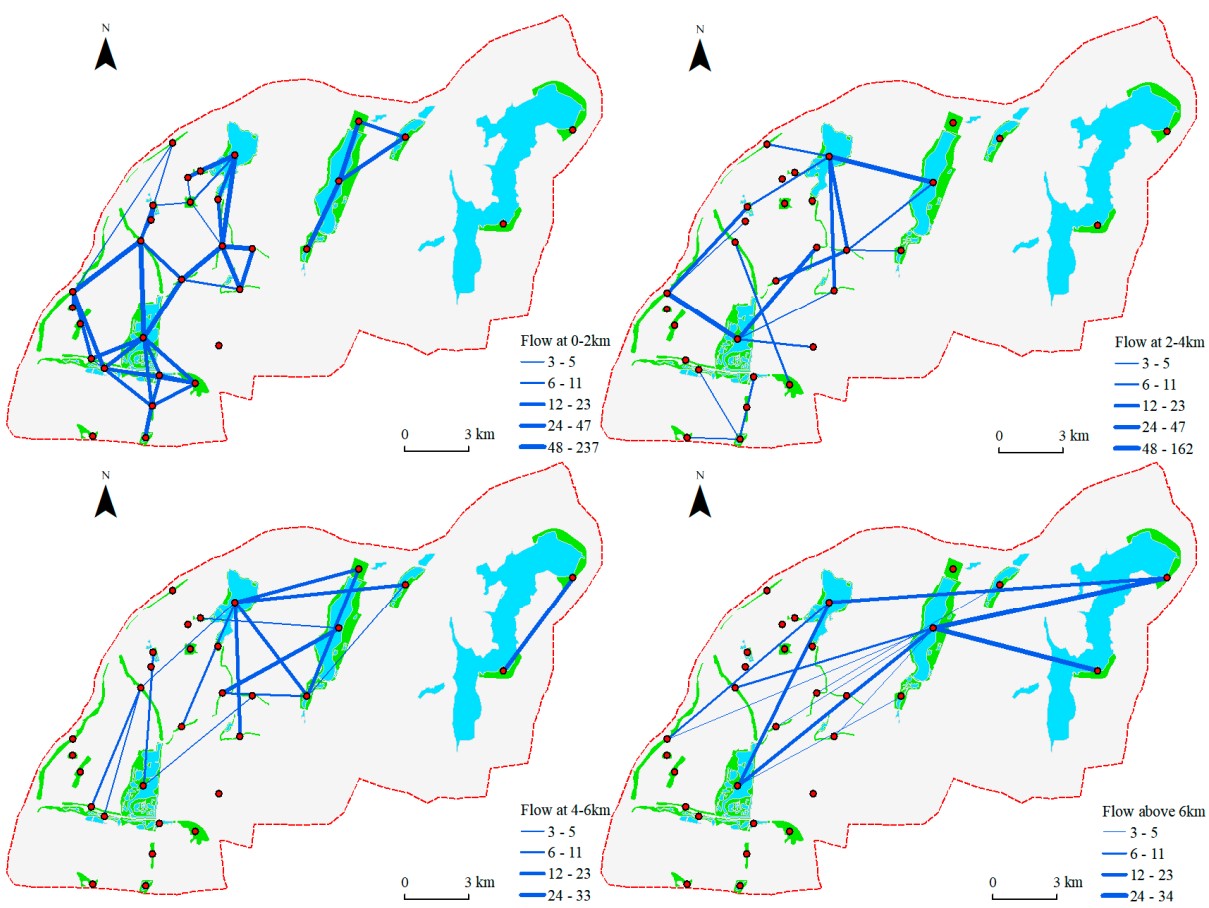

**Figure 6.** Spatial characteristics of the network under different distances.

Moreover, this study calculated the proportion of the flow of the different types of exercise under the given distances. It was found that people performed different types of exercise at different distances. With the increase in distance, the proportion of the walking and running flow decreased, while that of the cycling flow gradually increased. The running flow accounted for the largest proportion at 0–2 km (61.39%), but when the distance increased to 2–4 km, the cycling flow accounted for more than half of the total flow. When the distance was greater than 6 km, the cycling flow dominated.

### 3.2.4. The Status of First-Level Patches in the Flow Network

This study discussed the status of the first-level patches based on the outflow and inflow of exercisers. We extracted paths related to first-level patches with an outflow and inflow greater than 3. We also calculated the number of nodes, the proportion of the flow, and the average path flow. Equations (1) and (2) were used to calculate the weighted average distance of the flow between nodes. The results are shown in Figure 7 and Table 5.

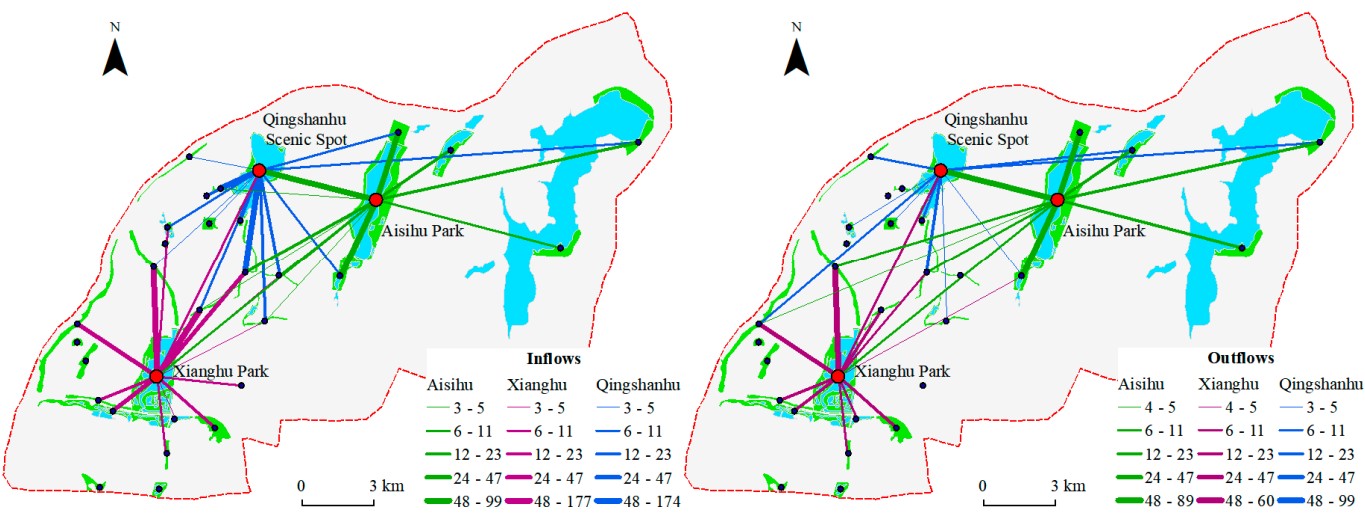

**Figure 7.** The spatial relationship between first-level patches and nodes.

**Table 5.** Flow characteristics of first-level patches.

| Patch Name | Qingshanhu Scenic Spot | | Xianghu Park | | Aisihu Park | |
|---|---|---|---|---|---|---|
| | Outflow | Inflow | Outflow | Inflow | Outflow | Inflow |
| Number of nodes | 12 | 16 | 12 | 15 | 10 | 12 |
| Proportion of the flow | 7.65% | 16.16% | 8.38% | 16.20% | 13.17% | 13.54% |
| Average flow of a path | 15.58 | 24.69 | 17.08 | 26.40 | 32.20 | 27.58 |
| Weighted average distance (km) | 3.77 | 1.68 | 1.86 | 1.34 | 2.81 | 2.60 |

Firstly, first-level patches have a large number of interconnected nodes and a large inflow and outflow, which were dominant in the whole flow. This study found 28 paths connected to Qingshanhu Scenic Spot and 27 paths connected to Xianghu Park. The inflow of the two nodes was larger than the outflow, which means they show agglomeration characteristics. Aisihu Park had 22 paths connected to it. In this park, little difference was found between the outflow and inflow. The proportions of the outflow and inflow of Aisihu Park were higher than 13%, meaning that the nodes are balanced. Generally speaking, the inflow and the proportion of the total flow of Qingshanhu Scenic Spot and Xianghu Park was larger than that of Aisihu Park. The former benefits from the corridors around them, especially Yudaihezong Channel and Fuhe Park, which are first-level corridors with a large flow. The three nodes were superimposed on one another, while duplicate ones were excluded, amounting to 65 paths. The outflow accounted for 29.20% of the total flow, while the inflow accounted for 45.89%, showing the central position of the three nodes in the flow network.

Secondly, weighted distance showed how dominant first-level patches were in the flow network. It was proven that the greater the weighted distance, the greater the influence of the node on the regional flow of green exercise. Because it is located in the city center, the Qingshanhu Scenic Spot had the highest weighted average distance based on the outflow (3.77 km), making it the largest scattered center for green exercise. Next, the weighted average distance of the inflow and outflow of Aisihu Park was greater than 2.5 km. This means that it is far from the city center and first-level patches and corridors, so a person needs to go a long way to reach Aisihu Park through first-level nodes. The weighted average distance between the inflow and outflow of Xianghu Park was relatively low (both less than 1.9 km). This result is related to the park's adjacent patches and corridors, meaning that the flow of green exercise mainly occurs between adjacent nodes.

*3.3. Flow Mechanism, Pattern, and Spatial Spillover Effect*

3.3.1. Analysis of the Mechanism behind the Flow of Green Exercise

In this study, the multiple linear regression model (Equation (4)) was used to fit the running and cycling flows (Table 6). The results showed that each explanatory variable passed the significance test (*t*-test) and did not show multiple collinearity (VIF value < 2). After adjustment, the $R^2$ value was more than 0.35, i.e., $R^2$ of the cycling flow was greater than 0.5. Overall, the selection of variables was appropriate. According to the regression coefficients of variables under the two types of exercise and their positive and negative effects, the following conclusions can be drawn.

**Table 6.** Multiple linear regression fitting of the cycling and running flows.

| Exercise Type | Variable | Unstandardized Coefficient | | Standardization Coefficient | *t* | Sig. | Collinear Statistics | |
|---|---|---|---|---|---|---|---|---|
| | | B | Standard Error | Beta | | | Tolerance | VIF |
| Cycling flow $R^2 = 0.504$ | Constant term | 0.522 | 0.254 | | 2.05 | 0.04 | | |
| | Route length of the outflow | 0.596 | 0.099 | 0.543 | 6 | 0 | 0.915 | 1.092 |
| | Route length of the inflow | 0.507 | 0.094 | 0.492 | 5.37 | 0 | 0.896 | 1.116 |
| | Distance | −0.334 | 0.056 | −0.544 | −5.9 | 0 | 0.889 | 1.125 |
| Running flow $R^2 = 0.358$ | Constant term | 1.637 | 0.225 | | 7.29 | 0 | | |
| | Route length of the outflow | 0.269 | 0.1 | 0.247 | 2.69 | 0.01 | 0.952 | 1.05 |
| | Route length of the inflow | 0.222 | 0.101 | 0.204 | 2.19 | 0.03 | 0.928 | 1.078 |
| | Distance | −0.481 | 0.075 | −0.597 | −6.4 | 0 | 0.935 | 1.069 |

(1) Firstly, regression coefficients of the length of the main exercise route in the outflow and inflow were greater than 0, indicating that length had a positive effect on the flow. Secondly, the regression coefficient of the length of the route in the outflow was slightly larger than that in the inflow. In other words, when the length of the exercise route of the adjacent green space was larger, it was easier for people to move between nodes. In addition, spatial distance between nodes was negative, which affected the flow of green exercise.

(2) According to the gravity model, there were spatial differences in the interpretation of the flow under different types of green exercise. The regression coefficient of the spatial distance of the running flow was −0.481, while that of the cycling flow was −0.334, indicating that the spatial distance constraint of the running flow was greater than that of the cycling flow. Likewise, the runners' demand for spaces closer to one another was higher. In addition, the regression coefficient of the length of the exercise route in the cycling flow (more than 0.5) was significantly larger than that in the running flow (about 0.36), indicating that the longer the exercise route, the stronger the attraction to cycling. In contrast, route length imposed less constraint on the running flow.

3.3.2. Analysis of Flow Patterns

When $I_{ij} \geq 12$, it was found that three agglomeration maps formed in the study area. Furthermore, this study considered $I_{ij} \geq 24$ as the main and $12 \leq I_{ij} < 24$ as the secondary flow. Combined with node hierarchy and flow characteristics, the whole flow network was divided into three subnetworks: Qingshanhu, Xianghu, and Aisihu, with first-level patches at their centers (Figure 8a). In the actual flow, subnetworks were relatively independent. However, the relationship between subnetworks was still maintained through the main flow path. Within a subnetwork, there was a close relationship between nodes. Particularly, the first-level and the second-level nodes were mainly connected through the main flow, while the third-level nodes were connected to other nodes through the secondary flow. Based on these findings, this study drew a spatial diagram of the flow between nodes in the subnetworks (Figure 8b).

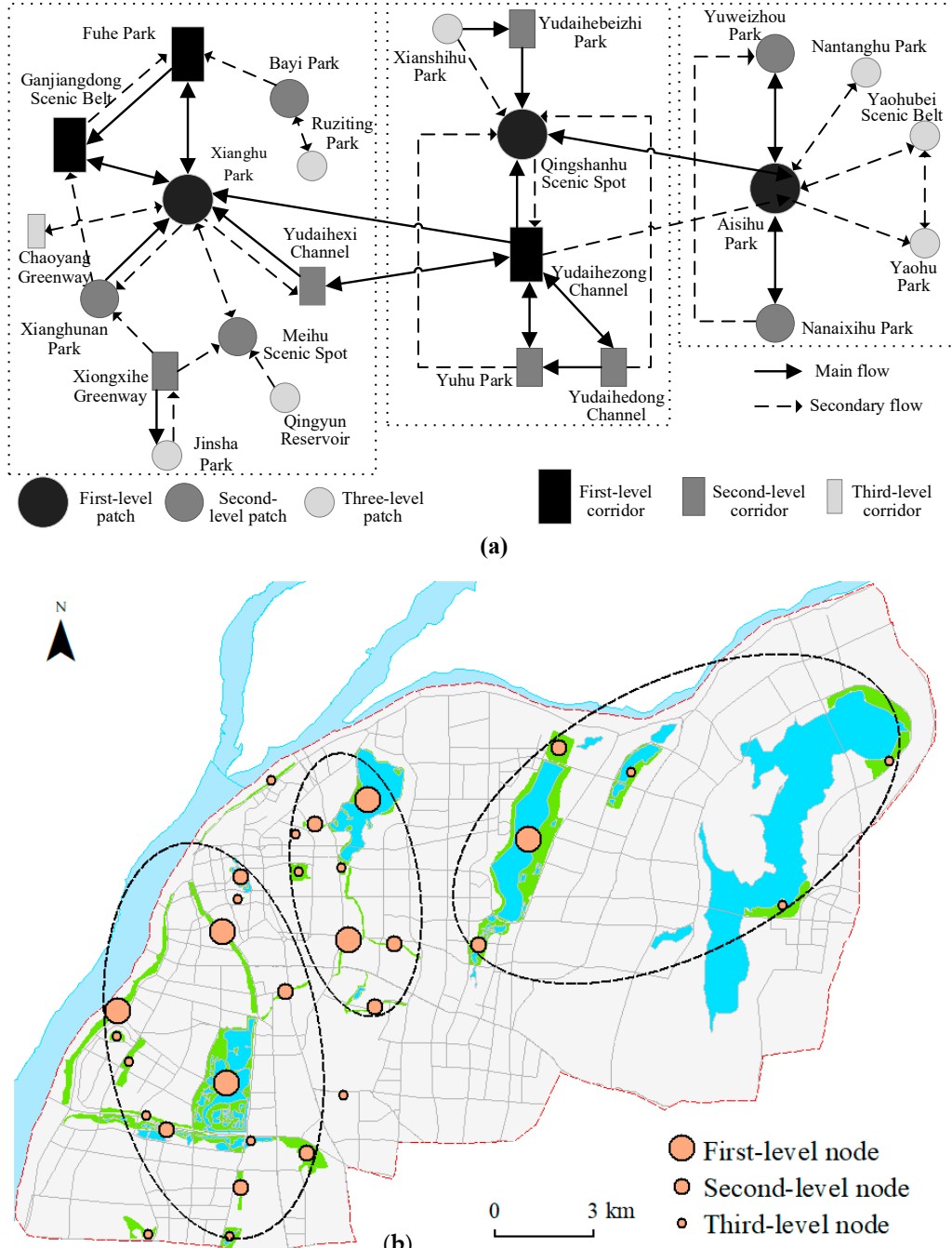

**Figure 8.** The division of the subnetworks in Nanchang: (**a**) the division of the flow network between the nodes; and (**b**) the location of the subnetworks. The dashed circle represents the approximate extent of the subnetwork.

Based on the analysis of the internal flow characteristics of each subnetwork, it was found that the Qingshanhu subnetwork had the Qingshanhu Scenic Spot as the central agglomeration node and that the subnetwork mainly directed a person to the Qingshan Scenic Spot through the corridor node. Moreover, the Xianghu subnetwork had Xianghu Park as the central agglomeration node. There, a person moved through patches and corridors. The Aisihu subnetwork had a balanced node, Aisihu Park, as its center, in which a person moved between the patches of the subnetwork. According to these characteristics, the green space network was divided into three types of flow patterns: corridor diffusion-center agglomeration of the Qingshanhu subnetwork, the patch and corridor

complementary center agglomeration of the Xianghu subnetwork, and the patch diffusion center balance of the Aisihu subnetwork. It can be seen from the three flow patterns that the central node (first-level patch node) plays a central organizational role in the agglomeration and diffusion of green exercisers. Thus, the construction of a central node is conducive to promoting the green exercise flow network. In addition, the higher the node level, the better the connection between the nodes in the network (Figure 8a). This can be manifested in a few ways. A first-level node develops a flow relationship with at least three nodes, a second-level node is connected to at least two nodes, while a third-level node is connected to only one node.

### 3.3.3. The Spillover Effect of the Urban Green Space Network

Distance was an important factor in assessing the attractiveness and accessibility of green spaces. When there is an urban green space network, one can ask the question: can it create a shortcut in a high-quality green space with long routes, promote the attractiveness of green spaces, and urge people to do activities on the long route?

Compared to type B samples, the standard ellipse ranges of Xianghu Park, Qingshan Scenic Spot, and Aisihu Park were expanded by 8.49 km$^2$, 5.42 km$^2$, and 7.15 km$^2$ in all sample data, respectively (Figure 9). This shows that with the support of the urban green space network, the spatial scope of exercise services of first-level patches was expanded, and their agglomeration was enhanced. The starting point of the exercisers in the Qingshan Scenic Spot mainly expanded to the south, about 0.83 km. Corridor nodes in the south, such as the Yudaihedong and the Yudaihedong channel, played an important role in diverting the exercisers to the Qingshan Scenic Spot. In Xianghu Park, exercisers were mainly directed to the northeast because first- and second-level corridors provided a large number of exercisers, thus expanding about 1.8 km to this direction. At the same time, the ellipse was also expanded by about 0.22 km to the south since Xianghunan Park and Xiongxihe Greenway directed some exercisers to the Xianghu Park. The main expansion direction of Aisihu Park was the southwest, about 0.86 km, due to Nanaixihu Park in the south and the Yudaihedong Channel, Yudaihedong Channel, and Qingshan Scenic Spot in the west.

Previous studies have devoted most of their attention to the ecological value of the urban green space network, i.e., animal migration [36] and biodiversity [37], while the equally important cultural service was ignored, including the exercise service value [38]. This kind of service value is realized through the flow of green exercise. An urban green space network provides a safe and green environment for exercising and enables people to reap great benefits from it. As can be seen from Figure 9, the urban green space network increased the distance over which people exercised and made them spend more time exercising. Thus, these people certainly reaped some health benefits. Intermediary corridors and patches became places that enabled people to reach their end points. Through these nodes, exercisers can enjoy the green spaces and exercise by taking longer time and a longer route to walk, run, or cycle.

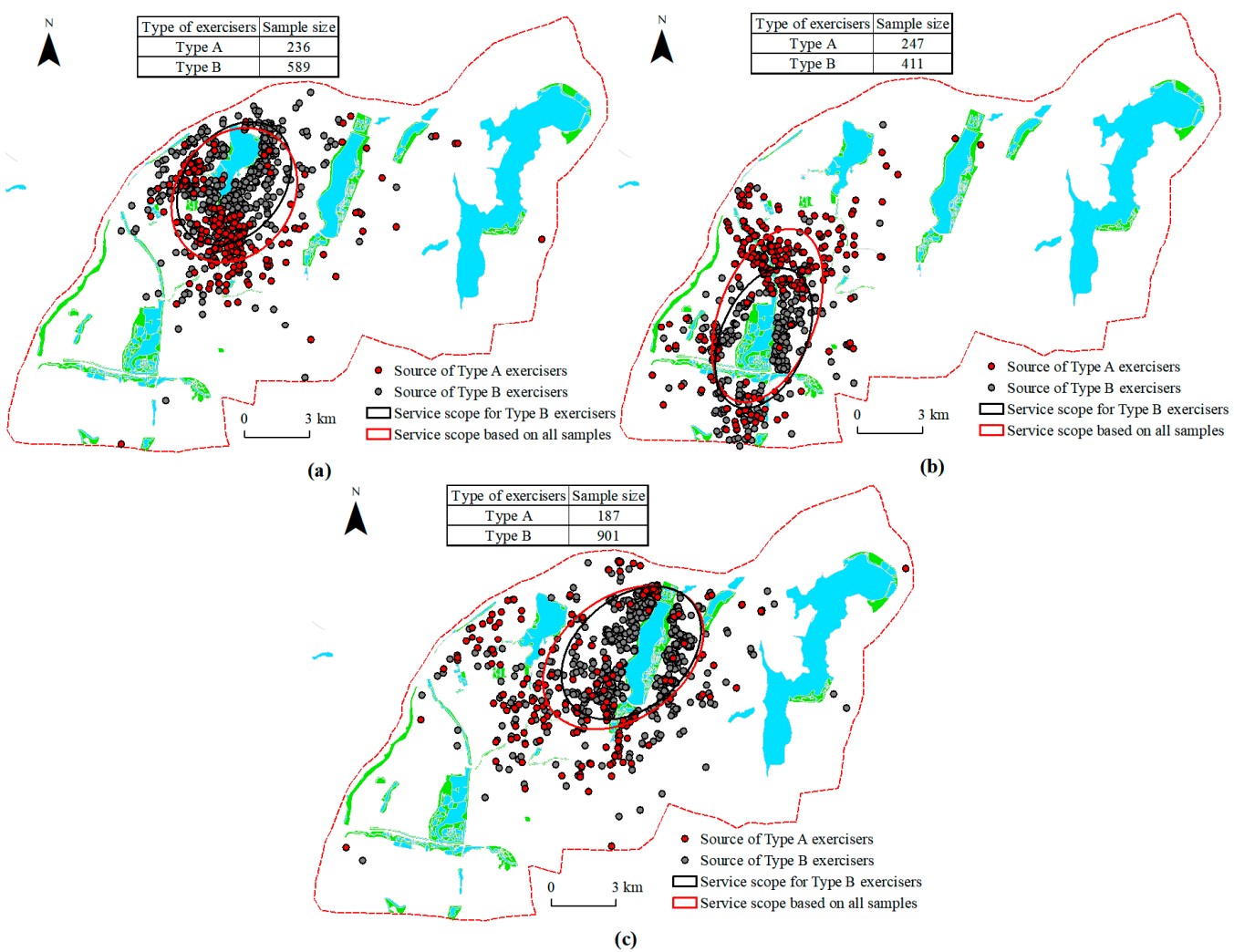

**Figure 9.** Source and distribution of different types of exercisers in the first-level patches: (**a**) Qingshan Scenic Spot; (**b**) Xianghu Park; and (**c**) Aisihu Park.

## 4. Discussion

### 4.1. The Optimization Path for the Urban Green Space Network

This study proposes the construction of an urban green space network (Figure 10) to enhance the flow of green exercise. To enhance the flow, node density also needs to be enhanced. A certain number of nodes enables the flow of green exercise. Increasing the number of nodes increases the number of paths and the flow, resulting in an indirect flow and a medium- to short-distance exercise route. In addition, the smaller the distance, the greater the flow. The layout of new park green spaces near already established nodes can generate more flow (Figure 10b). At present, there are many means of using the available space to build green spaces or park groups in the city or the suburbs.

Secondly, patch level should be improved. The flow between the green spaces is related to the level of its patches. A first-level patch is an important driver of the regional flow. Therefore, when considering the construction of an urban green space network, the construction of central nodes should be encouraged (Figure 10c). Possible measures include expanding the area of an existing green space or improving its quality (increasing the length of the exercise routes) to have more attractive spaces. Because the suburbs have more unoccupied land, new and large patches can be built there to promote the flow of green exercise.

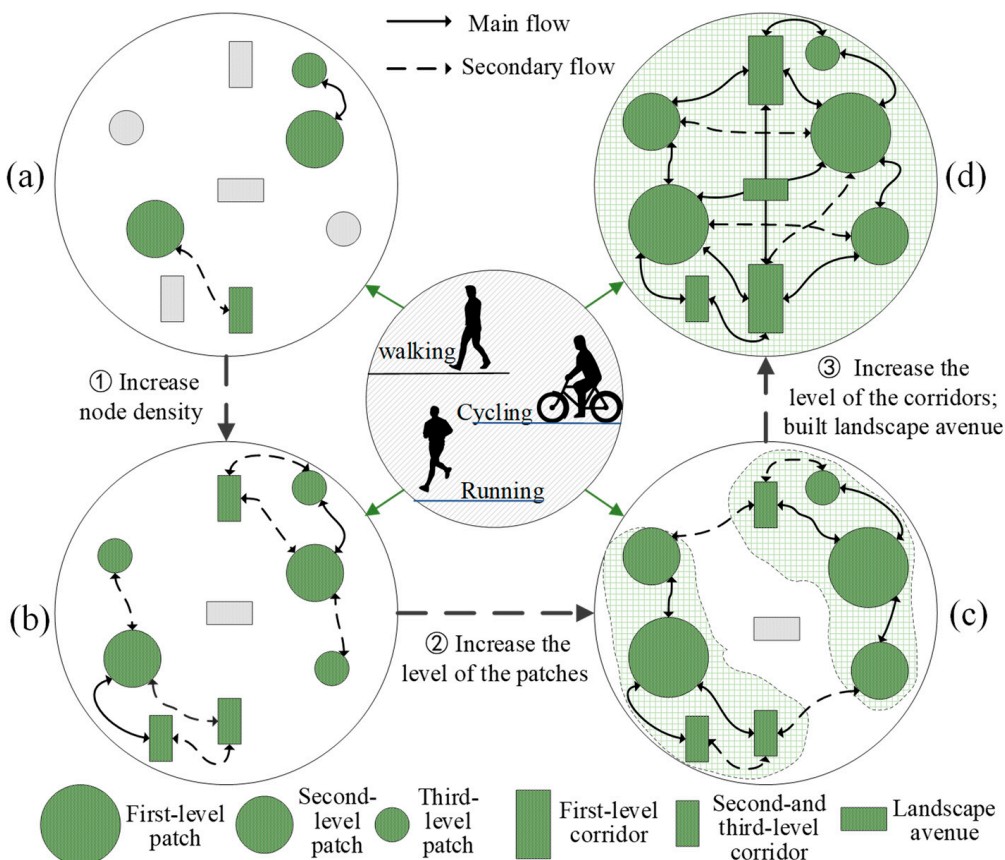

**Figure 10.** Optimization of the urban green space network based on the flow of green exercise (grey represents future green spaces, green represents built spaces). (**a**) Sparse connections between nodes; (**b**) Increased connections between first-level patches and their surrounding nodes; (**c**) Formation of subnetworks; (**d**) Formation of total flow network.

Lastly, the level of corridors should be improved, and new exercise routes should be built. Corridors play an important role in moving people from one patch to another. The significance of corridors is more obvious in southern China where blue and green infrastructure is better [39]. Therefore, this study suggests improving the abandoned or low-quality blue corridors in the city. In this study, several routes (landscape avenues) for running and cycling were built around Aisihu Park, which promoted the flow between Aisihu Park and other more distant patches. When the land in the city center is insufficient for more green spaces, the roads between the main nodes should be used to improve the flow between these nodes.

### 4.2. Limitations and Further Research

This study has some limitations. Although it combines the social network analysis method and the gravity model, the analysis cannot solve all problems related to the creation of a flow network. Due to a limited number of pages, results based on the two methods are not detailed enough. In addition, no data or analytical method is perfect. One of the limitations is the age-range of the exercisers who use the five applications. People who are tech-savvy, well-educated, and, most importantly, young fall into a specific age group.

Therefore, these results mainly reflect the behavior of the younger population. Attributes, components, and facilities of green spaces (accessibility, size, flora, water, sensory characteristics, microclimate, and amenities) have a greater influence on the willingness of the younger population to use green spaces and promote health [40], which ultimately makes some places more attractive than others. To better explain the driving mechanism behind the generation and evolution of the flow network, future research should combine

these methods of this study with other survey methods to analyze the movement of other types of exercisers, including groups who do not use fitness applications. Furthermore, this study collected the data of a user exercising on a certain route only once. That is to say, the sample data cannot reveal the temporal characteristics of the flow of green exercise. Likewise, the robustness of the results was affected by the scale of the data. Considering the impact of COVID-19 on residents' outdoor activities during 2020–2022 [41] and the local restriction of spatial mobility enforced through lockdown [42], the changes in green exercisers' mobility during the pandemic period will be discussed in the future. To draw more accurate conclusions, data collection of exercise routes should be improved. Considering that the attractiveness of green spaces is a relatively important factor in developing the flow of green exercise (only exercise route length was considered in this study) and that big data and machine learning have an important role in evaluating this attractiveness [43,44], future research should use relevant methods to understand how promoting attractiveness of green spaces affects the flow of green exercise.

## 5. Conclusions

This study examined the mobility of exercisers in the urban green space network in Nanchang. Although it is important to understand the mobility of exercisers, no empirical research has been conducted in this field. Using practical and available data from fitness applications, this study investigated the spatial characteristics of nodes in the network and the mechanism and pattern of the flow of green exercise using a variety of geospatial analysis techniques, such as social network analysis, the gravity model, and GIS-based mapping. These methods can comprehensively measure the mobility of exercisers in the urban green space network. The results show that there are differences in the hierarchy and attractiveness of nodes in the indirect flow of the flow network. It was found that first-level patches dominated the flow network. Likewise, the network changed due to the change in the exercise type, flow volume, and distance between nodes. The mechanism behind the flow was analyzed with the help of the traditional spatial interaction model. This study divided the nodes into three subnetworks, which were differentiated based on the type of flow in first-level patches and the patches' relationship with nodes. The urban green space network is required for the expansion of the service scope of first-level patches. Likewise, it can enhance the intensity of exercise. The main goal of the study was to optimize the network and enhance the flow of green exercise. The results changed the spatial view on green exercise and enriched the understanding of the social value of the urban green space network. Hopefully, this study will stimulate researchers and managers to pay more attention to the urban green space network and the mobility of exercisers.

**Author Contributions:** Z.C. wrote this article; D.G. embellished the language of the article; X.X. and L.Z. completed part of the data collection and processing work; C.F. reviewed the whole paper and put forward suggestions for improvement. All authors have read and agreed to the published version of the manuscript.

**Funding:** This research did not receive any specific grant from funding agencies in the public, commercial, or not-for-profit sectors.

**Data Availability Statement:** Not applicable.

**Acknowledgments:** The authors would like to acknowledge all the reviewers and editors.

**Conflicts of Interest:** The authors declare no conflict of interest.

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
