# Peer review of "The Flow of Green Exercise, Its Characteristics, Mechanism, and Pattern in Urban Green Space Networks: A Case Study of Nangchang, China"

_land, doi:10.3390/land12030673_

Round 1

Reviewer 1 Report

The authors present the characteristics of human flows among the parks of Nanchang city using green exercise data. The biggest problem with the study is that it is poorly formulated in some sections. The best example is the abstract. It uses concepts from social network analysis (such as node, first-level node, etc.) that are only defined in the methodology and results sections. (Later in the text, there are patch nodes, corridor nodes second and third-level nodes, sometimes, it is hard to follow.) Due to this problem, the presentation of the results in the abstract is incomprehensible. If the authors want to receive citations, it would be useful to include generally formulated results from the study, such as: distance had an important influence on green exercise, walkers move between adjacent parks, while runners move on longer distances and visit more parks and corridors, while cyclists cover the most extended distances and exploring the most green spaces. These types of results are understandable without reading the whole paper and can lead someone to cite it.

Furthermore, the potential readers of this paper (urban planners, and researchers of UGS) may not be familiar with social network analysis. So, the authors have to use its language and terms carefully. Finally, there are methods in urban planning to carry out similar analyses, such as space syntax. Only the last sentence of the abstract makes sense without reading the whole article.

 The same problem is present with the research questions. They should be formulated in a way that does not rely on technical terms only defined later. They are hard to understand and not clear. What do you mean by the term ‘spatial mode’? The determination of research objectives after the questions is entirely understandable. These are frustrating mistakes because the study applies a novel approach, and the results may be interesting for a wide readership.

The figures are of good quality and illustrate the authors' message well. The methodology is adequately developed, but instead of gathering everything in the 2.3 Research Methods subsection, there are scattered methodological texts in the various result subsections, e.g., line 222, lines 339-342 and 403-406.

The most exciting result for me was the determination of subnetworks based on the methodology (Fig. 7 a and 7b), which I would highlight much more than the authors.

Another problem is that section 4.1 does not belong to the Discussion section, as it presents new results.

Suggestions to the authors:

·      -revise the abstract,

·       -use terms in the introduction that have already been defined,

·    -provide a complete description of the methods and software used in the methodology section,

·    -place section 4.1 among the results, as it determines the spillover effect of the parks,

·   -finally, indicate in the title that this is a case study, as in its current form, it is misleading.

Author Response

Thanks a lot for your time and review of our manuscript. These comments are very helpful in improving the expression and content of the manuscript.

The original comments are cited in bold font (serial numbers for overall comments or specific comments are added according to reviewers’ comments), and all revised sections of the manuscript are indicated in red (including suggestions from other reviewers) with new line numbers added.

Reviewer 2 Report

As the authors stated in the discussion section; while previous studies have mostly focused on the ecological value of the urban green space network based on animal migration and biodiversity, the equally important cultural service, including the value of exercise service, has been overlooked. The existence of an urban green space network provides a safe and green environment to exercise and will allow people to derive great benefits from it. The importance of this study is remarkable in this headline.

The work is original and acceptable. Some improvements or suggestions for clarification are given below.

• “Nodes with a centrality greater than 160 represent the first level. This level includes three 195 patches and three corridor nodes, which are the most important areas for green exercising. 196 The second level constitutes five patches and five corridor nodes based on a centrality 197 between 60 and 160.” How it was determined should be stated.

• In the context of research on the spatial relationship between green spaces; By paying attention to the social value of the urban green space network, a hierarchical and interconnected green space suggestions can be given and concretized for the practitioners. It can be suggested how the factors affecting green exercise should be in the urban green space typology, for example, in urban parks or ecological corridors.

• Statistical evaluation assumptions and techniques of data related to walking, running and cycling should be expressed.

Author Response

(The authors gave the same response as above.)

Reviewer 3 Report

Dear authors,

Please find in the attached file my comments to this manuscript.

Thank you.

Author Response

(The authors gave the same response as above.)

Reviewer 4 Report

This paper is written well and objectives are very clear methods used in the results are resented is good manners, but some changes are required

1- The title need to be improve, remove the word “And” as it used in two time.

2-The abstract need revision. Author should shorter this section only explain the summary of the paper rather than the literature, Try to fit in 200 words only according to temple by LAND-MDPI.

3- retitle the section 2 section heading, like methods and split into study area sand methods.

4-  For patch-corridor-matrix, total 31 green spaces were selected, author should describe the criteria how to select these spaces.

5- In methodology section number of nodes for walking, running and cycling are 29,31,28 while total flow is 31, is it clear the total flow is 31 or something else?

6- For regression coefficient the R2 values are less than 0.3 for running flow can the author/authors explain the reason for it?

7- References: authors use some old references like 1st citation 1999, author should use update data/ research results.

8- Add more references to authenticate the results and research.

 This paper is written well and objectives are very clear methods used in the results are resented is good manners, but some changes are required

1- The title need to be improve, remove the word “And” as it used in two time.

2-The abstract need revision. Author should shorter this section only explain the summary of the paper rather than the literature, Try to fit in 200 words only according to temple by LAND-MDPI.

3- retitle the section 2 section heading, like methods and split into study area sand methods.

4-  For patch-corridor-matrix, total 31 green spaces were selected, author should describe the criteria how to select these spaces.

5- In methodology section number of nodes for walking, running and cycling are 29,31,28 while total flow is 31, is it clear the total flow is 31 or something else?

6- For regression coefficient the R2 values are less than 0.3 for running flow can the author/authors explain the reason for it?

7- References: authors use some old references like 1st citation 1999, author should use update data/ research results.

8- Add more references to authenticate the results and research.

Author Response

(The authors gave the same response as above.)
